# Relationship between Ideal Cardiovascular Health and Incident Proteinuria: A 5 Year Retrospective Cohort Study

**DOI:** 10.3390/nu14194040

**Published:** 2022-09-28

**Authors:** Yu-Min He, Wei-Liang Chen, Tung-Wei Kao, Li-Wei Wu, Hui-Fang Yang, Tao-Chun Peng

**Affiliations:** 1Division of Family Medicine, Department of Family and Community Medicine, Tri-Service General Hospital, National Defense Medical Center, Taipei 11490, Taiwan; 2Division of Geriatric Medicine, Department of Family and Community Medicine, Tri-Service General Hospital, National Defense Medical Center, Taipei 11490, Taiwan; 3Graduate Institute of Medical Sciences, National Defense Medical Center, Taipei 11490, Taiwan; 4Graduate Institute of Clinical Medicine, College of Medicine, National Taiwan University, Taipei 11490, Taiwan

**Keywords:** proteinuria, cardiovascular diseases, risk factors

## Abstract

The objective of this study was to examine whether a higher number of ideal cardiovascular health (CVH) metrics are beneficial for lowering the risk of proteinuria. This is a retrospective cohort study with an average follow-up of 5 years. Participants between 21 and 75 years old and without a history of cardiovascular disease and proteinuria were enrolled. CVH metrics, including smoking, diet, physical activity, blood pressure, body mass index (BMI), cholesterol, and fasting glucose, were assessed by questionnaires, physical examination, and blood analysis. Proteinuria was assessed by dipstick measurement. During the follow-up period, 169,366 participants were enrolled, and 1481 subjects developed proteinuria. A higher number of ideal CVH metrics was related to a lower risk of proteinuria after adjustment. Among the components of CVH metrics, ideal blood pressure (HR = 0.33, 95% CI = 0.25–0.43), fasting glucose (HR = 0.17, 95% CI = 0.12–0.22), and BMI (HR = 0.20, 95% CI = 0.15–0.27) had beneficial effects on proteinuria. Despite no significant benefit of diet score, the corresponding lower sodium intake showed a lower risk of proteinuria (HR = 0.58, 95% CI = 0.43–0.79). Incident proteinuria was inversely related to the number of ideal CVH metrics. CVH metrics may be a predictor of proteinuria, and achieving a higher number of ideal scores should be recommended as a proteinuria prevention strategy.

## 1. Introduction

Proteinuria, defined as urine protein excretion over 150 mg within 24 h, is characteristic of glomerular barrier dysfunction and may indicate kidney injury [1,2]. Although often benign and temporary, proteinuria is considered an early warning sign and confers a higher risk of chronic kidney disease (CKD), kidney failure with replacement therapy (KFRT), and even cardiovascular disease (CVD) [2,3,4,5]. Vascular endothelial dysfunction or an increased thrombosis risk may be a possible explanation [6,7]. Due to the strong relationship between proteinuria and cardiovascular disease, proteinuria is thought to be an independent predictor of cardiovascular diseases such as blood pressure and cholesterol [2,5,8]. Due to their important activities, some medications for hypertension and diabetes, such as angiotensin-converting-enzyme inhibitors (ACEIs), angiotensin receptor blockers (ARBs), and sodium glucose cotransporter 2 (SGLT2) inhibitors, have been shown to reduce proteinuria [9,10]. However, in addition to medication therapy, preventive strategies for proteinuria are critically important. Previous studies have shown that some metabolic and lifestyle factors, such as hypertension, obesity, smoking, and a high sodium diet, may increase the risk of proteinuria [11,12,13,14,15]. These findings indicate that avoiding some risk factors and, further, achieving healthy lifestyle habits were helpful for reducing proteinuria.

Previous studies focused on the association of incident proteinuria with either lifestyle habits or metabolic profiles. Few studies have comprehensively examined both lifestyle habits and metabolic profiles at the same time, even though both are equally important for proteinuria. Cardiovascular health (CVH) metrics, developed in 2010 by the American Heart Association (AHA) with the aim of improving cardiovascular health, provide an exhaustive way to assess both lifestyle habits and metabolic profiles. The metrics consist of seven cardiovascular disease risk factors and lifestyle habits, including smoking status, diet, physical activity, blood pressure (BP), body mass index (BMI), total cholesterol level, and fasting glucose [16]. A higher number of ideal cardiovascular health metrics indicate a lower risk of CVD, diabetes, and KFRT [17,18,19]. The connection between cardiovascular health and proteinuria in clinical practice has not been well investigated. If achieving ideal cardiovascular health could reduce the risk of proteinuria, these metrics might be a good indicator for proteinuria prevention.

The purpose of this study was to examine whether having a higher number of ideal CVH metrics is beneficial for lowering the risk of proteinuria in the Taiwanese population.

## 2. Methods

### 2.1. Study Population and Design

Our study was a retrospective cohort study with an average follow-up of 5 years. All the data were collected from the MJ Health Management Institution between 2000 and 2016 [20]. The institution provides a comprehensive health database in Taiwan with data from health screening programmes, consisting of urine analyses, blood analyses, physical examinations, and health questionnaires. All the medical tests or questionnaires had standard protocols and the International Organization for Standardization (ISO) certification [21]. We included participants between the ages of 21 and 75 years who completed all the health questionnaires (including dietary pattern and lifestyle habits), physical examination (including BP measurement, body weight, and body height), blood analysis (including total cholesterol, serum glucose, and serum creatinine), and urine analysis (including dipstick test). To avoid the influence of cardiovascular disease and assess the primary prevention of proteinuria, subjects with a history of cardiovascular disease and proteinuria at baseline were excluded. Figure 1 presents the flowchart of the selection procedure. Among the 212,545 participants screened, 37,775 subjects without data on proteinuria or CVH metrics were excluded. Of the remaining participants, 3775 subjects with a previous history of cardiovascular disease and 1629 subjects with proteinuria at baseline were excluded. Ultimately, 169,366 eligible subjects were enrolled in our study. Participants agreed and signed the consent form, allowing anonymous private data to be applied only for research aims. This study was approved by the Institutional Review Board of the Tri-Service General Hospital, Taiwan.

### 2.2. Sociodemographic and Clinical Variables

Information on age, sex, family income, education level, and alcohol consumption were obtained from a self-reported questionnaire at baseline. The serum creatinine (Cr) level was measured by blood analysis with a standard protocol. Education level was divided into two groups according to the graduation of high school. Low education was defined as below or equivalent to high school and high education was defined as beyond high school. Family income was measured as annual income and used the unit of new Taiwan dollars (NTD). Low family income was below or equivalent to 1.2 million NTD and high family income was over 1.2 million NTD. Alcohol consumption was measured by drinking frequency per week based on the answers to the questionnaire. Participants were classified as nondrinkers, former drinkers, mild drinkers (1–2 times/week), moderate drinkers (3–4 times/week), or heavy drinkers (≥5 times/week), from lowest to highest frequency.

### 2.3. Proteinuria

Proteinuria was evaluated by urine dipstick, which was collected from random spot urine measurements. The Roche Miditron M semiautomated computer-assisted urinalysis system (Combur-10 test M dipstick, Basel, Switzerland) was applied to assess proteinuria [20]. The urine dipstick reading was defined as negative or trace for <30 mg/dL, 1+ for 30–99 mg/dL, 2+ for 100–299 mg/dL, 3+ for 300–999 mg/dL, and 4+ for ≥1000 mg/dL. In our study, we defined negative proteinuria as negative or trace proteinuria and positive proteinuria as at least 1+ on the dipstick test.

### 2.4. Cardiovascular Health Metrics

According to the AHA definition, the CVH metrics include smoking status, diet, physical activity, blood pressure, BMI, total cholesterol level, and fasting glucose [16]. Each component was categorized into ideal, intermediate, and poor. Table 1 showed the ideal, intermediate, and poor definition of each component of CVH metrics. Smoking status, which was obtained from self-report questionnaires, was defined as ideal, intermediate, and poor as never smoking, former smoking, and current smoking, respectively. The healthy diet score was evaluated according to a standardized semiquantitative food frequency questionnaire (FFQ) developed by MJ Health Management [21]. This was composed of 22 nonoverlapping questions that assessed the daily consumption of different food components based on data on intake frequency and portion size. The healthy diet score was determined by the intake of five different foods, including sodium (daily consumption < 1500 mg), fish (weekly consumption ≥ 198 g), fruits and vegetables (daily consumption ≥ 450 g), fiber-rich whole grains (daily consumption ≥ 85 g), and sugar-sweetened beverages (weekly consumption ≤ 1 L). Based on the AHA’s recommendation, it was categorized as the ideal for intake of 4–5 components, intermediate for intake of 2–3 components, and poor for intake of 0–1 components. Physical activity was calculated by self-report questionnaires that included the exercise frequency, duration, and intensity. We defined ideal physical activity level as exercise ≥ 210 min/week, intermediate as exercise 60–210 min/week, and poor as exercise < 60 min/week. Blood pressure (BP) was measured twice with the patient in a seated position after at least 5 min of rest. We categorized ideal BP as systolic blood pressure (SBP) < 120 mmHg and diastolic blood pressure (DBP) < 80 mmHg, intermediate BP as SBP 120–139 mmHg or DBP 80–89 mmHg, and poor BP as SBP ≥ 140 mmHg or DBP ≥ 90 mmHg. BMI was calculated based on measured body height and weight. BMI was classified as ideal, intermediate, and poor according to values <25 kg/m^2^, 25–29.9 kg/m^2^, and ≥30 kg/m^2^, respectively. To assess total cholesterol and fasting glucose, the subjects fasted for over 8 h before the collection of blood samples, and the samples were analyzed following a standard procedure. Total cholesterol was defined as ideal, intermediate, and poor based on the values <200 mg/dL, 200–239 mg/dL, and ≥240 mg/dL, respectively. Fasting glucose was defined as ideal, intermediate, and poor as the values <100 mg/dL, 100–125 mg/dL, and ≥126 mg/dL, respectively. A higher number of ideal status of CVH metrics indicated better cardiovascular health.

### 2.5. Statistical Analysis

Age, creatinine level, blood pressure, BMI, total cholesterol, and fasting glucose are categorized as continuous variables. The values are described as the mean and were analyzed with Student’s t-test. Sex, education level, family income, alcohol consumption, proteinuria, ideal smoking status, ideal healthy diet score, ideal physical activity, ideal blood pressure, ideal BMI, ideal total cholesterol, and ideal fasting glucose are classified as categorical variables. The values are described as frequencies and were analyzed with the chi-square test depending on the baseline data. The CVH metrics were considered independent variables, and incident proteinuria was a dependent variable. Cox regression analysis was applied to evaluate the relationship between CVH metrics and incident proteinuria. To assess the protective effect of proteinuria between different components of the CVH metrics and healthy diet score, the association of proteinuria with each component of the CVH metrics and healthy diet score were investigated by Cox regression analysis. There were two different multivariable adjustments in our study. Model 1 was adjusted for age and sex. Model 2 was adjusted for age, sex, education level, creatinine level, alcohol consumption, and family income. A two-tailed *p* value < 0.05 was significant in our study. Data analyses and calculations were performed by the Statistical Package for the Social Sciences version 22 (SPSS, Inc., Chicago, IL, USA).

## 3. Results

### 3.1. Baseline Characteristics

A total of 169,366 subjects were enrolled in the final analyses. Table 2 shows the baseline characteristics over different numbers of ideal CVH metrics. The mean age of all study participants was 39.3 years, and 53.8% were men. The average creatinine level was 0.96 mg/dL. The participants with higher CVH metrics were significantly younger and more likely to be female and had higher education levels but lower alcohol consumption and family income (*p* < 0.01). As defined, the patients with higher CVH metrics showed a tendency toward lower BMI, total cholesterol, fasting glucose, smoking status, and blood pressure (*p* < 0.01).

### 3.2. Proteinuria and Cardiovascular Health Metrics

Over a median of 5 years of follow-up, 1481 subjects developed proteinuria. The relationship of incident proteinuria with different CVH statuses and number of ideal CVH metrics are shown in Table 3. We divided the subjects with different numbers of ideal CVH metrics into three subgroups: low (0–2), moderate (3–4), and high (5–7) CVH status. A graded decrease in the hazard ratio of the presence of proteinuria was seen among participants with low (HR = 1, ref), moderate (HR = 0.39, 95% CI = 0.35–0.44), and high (HR = 0.20, 95% CI = 0.17–0.24) CVH status in the unadjusted model. In model 1, similar findings were observed among patients with low (HR = 1, ref), moderate (HR = 0.45, 95% CI = 0.40–0.50), and high (HR = 0.27, 95% CI = 0.23–0.32) CVH status. In model 2, compared to low CVH status, moderate and high CVH status had healthier rates of the presence of proteinuria (HR = 0.46 [95% CI = 0.37–0.57] and 0.41 [95% CI = 0.30–0.55], respectively).

Similarly, the hazard ratio of incident proteinuria decreased gradually as the number of ideal CVH metrics increased in the unadjusted model. The hazard ratios from the scale of 0 to 6–7 were 1 (ref), 0.64 (95% CI = 0.49–0.83), 0.40 (95% CI = 0.31–0.51), 0.26 (95% CI = 0.20–0.33), 0.15 (95% CI = 0.11–0.19), 0.10 (95% CI = 0.08–0.13), and 0.10 (95% CI = 0.06–0.15). An identical trend was seen in model 1. The hazard ratios were 1 (ref), 0.60 (95% CI = 0.46–0.78), 0.38 (95% CI = 0.29–0.49), 0.26 (95% CI = 0.20–0.34), 0.16 (95% CI = 0.13–0.21), 0.13 (95% CI = 0.09–0.17), and 0.10 (95% CI = 0.06–0.16) from the 0 to 6–7 scale. In model 2, higher CVH metrics were related to a lower presence of proteinuria. The hazard ratios were 1 (ref), 0.73 (95% CI = 0.45–1.17), 0.50 (95% CI = 0.31–0.80), 0.34 (95% CI = 0.21–0.54), 0.22 (95% CI = 0.13–0.36), 0.25 (95% CI = 0.15–0.42), and 0.12 (95% CI = 0.04–0.36) from the 0 to 6–7 scale.

### 3.3. Proteinuria and Each Component of CVH Metrics

The relationship between the presence of proteinuria and each component of the CVH metrics is shown in Figure 2. In the unadjusted model, compared to poor CVH metric scores, the risk of proteinuria was lower with ideal blood pressure (HR = 0.19, 95% CI = 0.17–0.22), ideal total cholesterol (HR = 0.46, 95% CI = 0.39–0.53), ideal fasting glucose (HR = 0.08, 95% CI = 0.07–0.10), ideal body mass index (HR = 0.15, 95% CI = 0.13–0.17), and ideal smoking status (HR = 0.71, 95% CI = 0.63–0.80). In contrast, the hazard ratio of incident proteinuria was higher with ideal physical activity (HR = 1.25, 95% CI = 1.08–1.46) and ideal healthy diet score (HR = 1.32, 95% CI = 1.09–1.60). In model 1, incident proteinuria was decreased with ideal blood pressure (HR = 0.29, 95% CI = 0.25–0.33), ideal total cholesterol (HR = 0.65, 95% CI = 0.56–0.76), ideal fasting glucose (HR = 0.13, 95% CI = 0.11–0.16), ideal body mass index (HR = 0.17, 95% CI = 0.14–0.20), ideal smoking status (HR = 0.72, 95% CI = 0.63–0.82), and ideal physical activity (HR = 0.74, 95% CI = 0.63–0.87) compared to poor CVH metric scores. There was no specific benefit of the ideal health diet score. In model 2, a similar benefit to incident proteinuria was seen for ideal blood pressure (HR = 0.33, 95% CI = 0.25–0.43), fasting glucose (HR = 0.17, 95% CI = 0.12–0.22), and body mass index (HR = 0.20, 95% CI = 0.15–0.27). There was no significant difference with ideal total cholesterol, smoking status, physical activity, or healthy diet score compared to poor CVH metric scores.

### 3.4. Proteinuria and CVH Healthy Diet Score

The association between incident proteinuria and each component of the CVH healthy diet score is presented in Table 4. Among the five components of the healthy diet score, only lower sodium intake showed a lower rate of incident proteinuria (HR = 0.76, 95% CI = 0.63–0.91) in the unadjusted model. Higher incident proteinuria was identified with higher whole grain (HR = 1.33, 95% CI = 1.05–1.69) and lower sugar-sweetened beverage consumption (HR = 1.24, 95% CI = 1.11–1.37). There was no significant difference in incident proteinuria regarding fruit and vegetable or fish intake. In model 1, lower sodium intake was associated with a lower risk of proteinuria (HR = 0.68, 95% CI = 0.57–0.82). No significant difference in incident proteinuria was seen in the fruit and vegetable, whole grain, fish, and sugar-sweetened beverage groups. Similarly, in model 2, only lower sodium intake showed a beneficial effect on incident proteinuria (HR = 0.58, 95% CI = 0.43–0.79). No difference in incident proteinuria was identified for the other four components.

## 4. Discussion

In this cohort study of the Taiwanese population, increased scores on the CVH metrics showed graded and beneficial effects for incident proteinuria, independent of age, sex, education level, creatinine level, family income, and alcohol consumption.

The association between CVH metrics and incident proteinuria provides a new strategy for reducing proteinuria. A reduction in proteinuria was found to be related to a protective effect against renal function decline and cardiovascular events [2,3,4,5,22]. To this end, some medications for hypertension and diabetes have shown a beneficial effect on proteinuria [9,10]. However, in addition to treatment, we should emphasize the prevention of proteinuria. Generally, the cause of proteinuria is thought to be glomerular barrier dysfunction. The risk factors associated with vascular endothelial damage or inflammation may cause injury to the glomerular filtration barrier and induce proteinuria [5,23]. Given the association of proteinuria with cardiovascular and metabolic risk profiles, some evidence has suggested that blood pressure, fasting glucose, BMI, smoking, and physical activity had possible effects on incident proteinuria in previous studies [11,12,13,15]. However, the results are inconsistent and lack organized evaluation. Rashidbeygi et al. compared different metabolic profiles and assessed the risk of proteinuria by metabolic syndrome. Metabolic syndrome and its components were found to be related to higher incident proteinuria [24]. Wakasugi et al. [25] and Okada et al. [26] conducted population-based cohort studies and showed that healthy lifestyle modification could lower the incidence of proteinuria. Our study combined both metabolic profiles and lifestyle habits and demonstrated a strong inverse relationship between higher CVH metric scores and the incidence of proteinuria. The CVH metrics could be a comprehensive predictor and useful for the prevention of proteinuria.

We evaluated the association between each component of the CVH metrics and incident proteinuria. After multivariable adjustment, three components, including ideal blood pressure, fasting glucose, and BMI, showed a significant effect on the presence of proteinuria. There were some explanations provided in previous studies. In hypertension, renal damage may be caused by direct pressure injury to the vascular bed. Long-term hypertension may induce dysfunction of autoregulatory vasoconstriction of the preglomerular vasculature and worsen kidney injury and proteinuria [27,28]. Regarding hyperglycemia, the UK Prospective Diabetes Study showed that poor control of blood sugar in type 2 diabetes was associated with higher renal-retinal complications [29]. Hyperglycemia-induced oxidative stress with overproduction of mitochondrial reactive oxygen species (ROS) may be a possible reason for vascular endothelial cell damage [30,31]. Poor BMI was related to obesity. Compensatory hyperfiltration occurs due to the raised metabolic demands of increased body weight [32,33]. Increased glomerular pressure, along with oxidative stress, inflammation, and renin–angiotensin–aldosterone system (RAAS) activation, causes endothelial dysfunction and increases the risk of proteinuria [34]. For the prevention of proteinuria, ideal blood pressure, fasting glucose, and BMI are critical.

The other nondietary components of the CVH metrics, including ideal total cholesterol, ideal smoking status, and ideal physical activity, did not show significantly better effects on incident proteinuria. The result of dyslipidemia was consistent with the Atherosclerosis Risk in Communities (ARIC) study and Chronic Renal Insufficiency Cohort (CRIC) study, which found that total cholesterol and LDL were not associated with renal disease progression [35,36]. In contrast, individuals with hypertriglyceridemia and lower HDL, which are components of metabolic syndrome, showed significant increases in proteinuria [24]. Cigarette smoking may increase ROS production and is harmful to the cardiovascular system [37]. However, the role of smoking on kidney health is controversial. Maeda et al. and Ito et al. showed that smoking increased the risks for new-onset CKD and proteinuria among Japanese individuals [38,39]. Conversely, a meta-analysis published by Xia et al. indicated that smoking increased the risk of CKD but had no association with proteinuria in the general adult population [40], which was compatible with our study. The ideal smoking status may have a mild benefit on kidney health but a limited effect on proteinuria [35]. We did not observe a relationship between physical activity and proteinuria. Some but not all observational studies suggested that regular physical activity was associated with lower incidence rates of CKD and proteinuria [41,42]. Long-term regular physical activity can improve vascular endothelial dysfunction and lower the incidence rates of obesity, hypertension, and diabetes [43]. The benefits of exercise for kidney injury and chronic kidney disease are reasonable. However, it was difficult to detect this effect in our study, and incorrect self-reports and a lack of information on the habitual duration of physical activity may be possible limitations. Although ideal total cholesterol, ideal smoking status, and ideal physical activity showed little or no better effect on proteinuria, they had a cumulative effect and were connected with kidney health.

For healthy diet, lower sodium intake was related to a lower risk of proteinuria, whereas the other four components and ideal healthy dietary score did not reduce proteinuria in our study. Higher sodium intake was found to be a risk factor for hypertension and was associated with an increased incidence of proteinuria and CKD progression in previous studies [44,45]. The adverse effect of salt intake on the kidneys may be associated with increased blood pressure and RAAS system dysregulation [46]. Previous studies indicated that dietary sodium restriction could potentiate the effect of RAAS blockade and decrease blood pressure, which is helpful for proteinuria [47,48]. Regarding the other four components of the healthy diet score, previous studies revealed that some healthy dietary patterns, such as higher consumption of whole grains, dairy, and nuts, were associated with a lower risk of CKD in the ARIC cohort study [49,50]. However, this trend was not observed for the healthy dietary score of the CVH metrics [35]. Some, but not all, components of the healthy diet score, such as lower sodium intake, had a preventive effect against incident proteinuria.

There were a few studies exploring the association between cardiovascular health and kidney disease previously. In Japan, Suzuki et al. conducted a cohort study to explore the association between cardiovascular health and proteinuria by the modified CVH metrics [51]. Different from the AHA recommendation and our article, Suzuki et al. defined the ideal physical activity as exercise 30 min at least twice a week, and ideal dietary habit as skipping breakfast <3 times per week. Similar to our study, they revealed that the higher CVH status was related to a lower risk of proteinuria after adjustment for age and gender. However, besides the ideal blood pressure, fasting glucose, and BMI, Suzuki et al. also revealed the ideal smoking status, physical activity, and dietary habit had a protective effect for proteinuria. A different definition of CVH metrics and adjusted methods may be possible explanations. Rebholz et al. showed an association between CVH metrics and CKD by analyzing the results of the ARIC cohort study [35]. Muntner et al. mentioned the role of CVH metrics in KFRT according to the population-based Reasons for Geographic and Racial Differences in Stroke (REGARDS) Study [19]. Consistent with proteinuria, a graded relationship between the numbers of ideal CVH metrics and the risk of CKD and KFRT was seen. Notably, although the severity of kidney injury was different, two components of CVH metrics, ideal blood pressure and ideal fasting glucose, showed a significant beneficial effect on proteinuria, CKD, and even KFRT. The importance of blood pressure and fasting glucose need to be highlighted among subjects with each stage of renal function. For the promotion of kidney health, achieving ideal CVH, especially blood pressure and fasting glucose, is reasonable.

Our study has some potential limitations that should be considered when interpreting these findings. First, we used a self-report questionnaire to assess partial ideal health behaviors, such as smoking status, healthy diet, and physical activity. Although the questionnaire created by the MJ group was standardized and certified by ISO, the bias from inaccurate self-reports may limit our ability to detect an association. Second, the habitual duration of health behaviors was not considered in the CVH metrics. One of the purposes of CVH metrics was to provide simple and accessible guidance on cardiovascular health to individuals. Thus, the AHA did not add the habitual duration of each behavior to the metrics. The benefit to proteinuria might be affected by the amount of time the health behaviors are maintained. Third, we did not consider the impact of medicine. Some medications for hypertension and diabetes, such as ACEI/ARB and SGLT2, could decrease the incidence of proteinuria [9,10]. This effect would mask hypertension- or hyperglycemia-related proteinuria and lead to potential bias in our analysis. In addition, bias may be due to the long sample collection period due to changes in lifestyle and environment. Further long-term, prospective, multinational studies are helpful for explaining the relationship between CVH metrics and incident proteinuria. The metrics could be revised and more suitable for the promotion of the kidney health in the future.

## 5. Conclusions

In conclusion, our cohort study demonstrated a graded inverse relationship between the numbers of ideal CVH metrics and incident proteinuria in a Taiwanese population. In particular, ideal blood pressure, ideal fasting glucose, ideal BMI, and lower sodium intake were found to be significantly protective against proteinuria. The CVH metrics may be a predictor of proteinuria. Achieving a high number of ideal metrics, keeping normal blood pressure, fasting glucose, BMI, and decreasing sodium intake should be recommended as a proteinuria prevention strategy.

## Figures and Tables

**Figure 1 nutrients-14-04040-f001:**
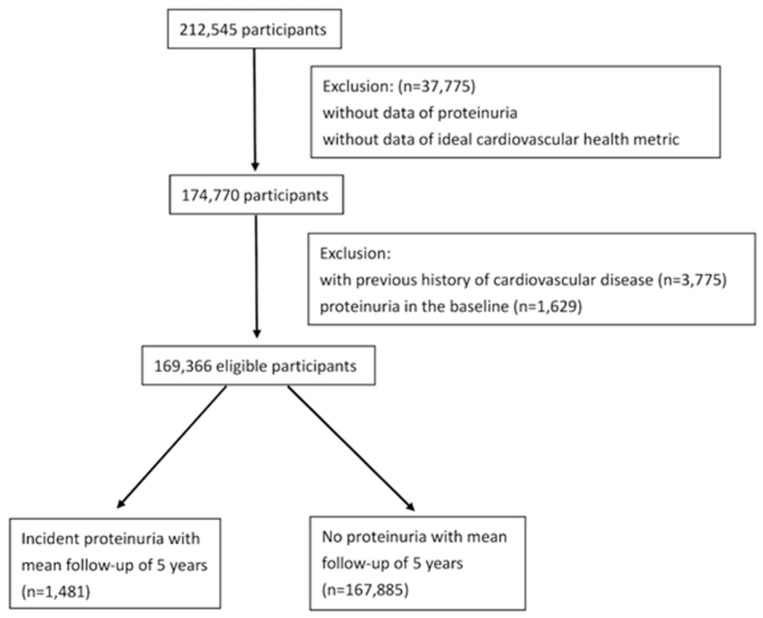
Flowchart of the selection procedure.

**Figure 2 nutrients-14-04040-f002:**
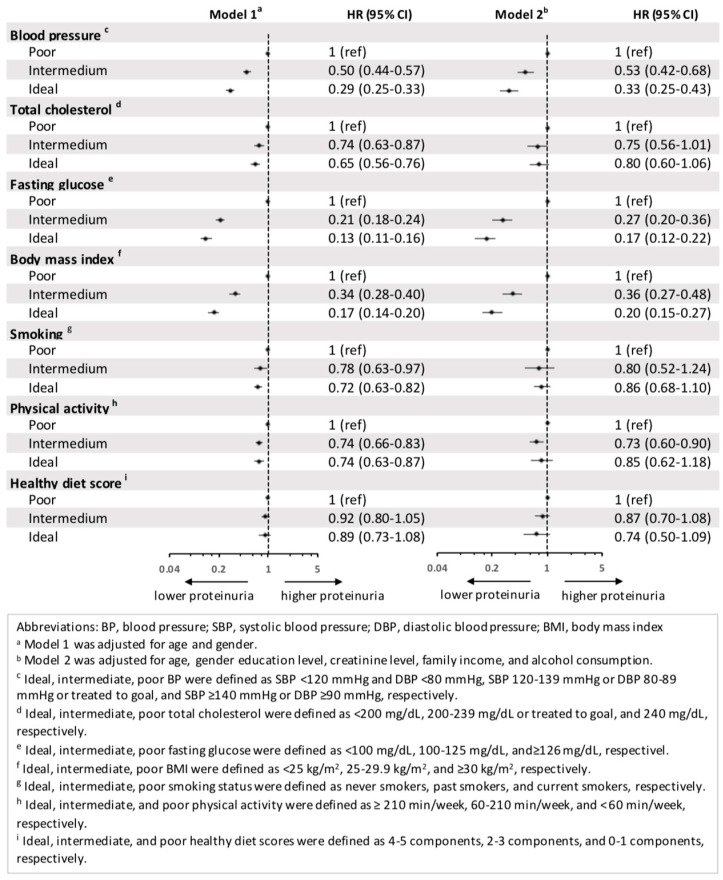
Hazard ratio for incident proteinuria between each component of cardiovascular health metrics.

**Table 1 nutrients-14-04040-t001:** The ideal, intermediate, and poor definition of each component of CVH metrics.

	Ideal	Intermediate	Poor
Smoking status	never smoking,	former smoking	current smoking
Healthy diet score ^a^	intake of 4–5 components	intake of 2–3 components	intake of 0–1 components
Physical activity	exercise ≥ 210 min/week	exercise 60–210 min/week	exercise < 60 min/week
Blood pressure	SBP < 120 mmHg and DBP < 80 mmHg	SBP 120–139 mmHg or DBP 80–89 mmHg	SBP ≥ 140 mmHg or DBP ≥ 90 mmHg
Body mass index	<25 kg/m^2^	25–29.9 kg/m^2^	≥30 kg/m^2^
Total cholesterol	<200 mg/dL	200–239 mg/dL	≥240 mg/dL
Fasting glucose	<100 mg/dL	100–125 mg/dL	≥126 mg/dL

^a^. The components of healthy diet score included sodium (daily consumption < 1500 mg), fish (weekly consumption ≥ 198 g), fruits and vegetables (daily consumption ≥ 450 g), fiber-rich whole grains (daily consumption ≥ 85 g), and sugar-sweetened beverages (weekly consumption ≤ 1 L).

**Table 2 nutrients-14-04040-t002:** Characteristics of the study participants.

	**Number of Ideal CVH Metrics**								
**Continuous Variables ^a^**	**All** **(*n* = 169,366)**	**0** **(*n* = 2539)**	**1** **(*n* = 11,768)**	**2** **(*n* = 25,391)**	**3** **(*n* = 38,674)**	**4** **(*n* = 47,044)**	**5** **(*n* = 38,706)**	**6–7** **(*n* = 5244)**	***p*-Value**
Age, mean (SD), years	39.3 (11.6)	42.8 (10.2)	43.8 (11.5)	43.3 (12.1)	41.1 (12.0)	37.8 (11.0)	35.2 (9.7)	39.1 (11.4)	<0.01
BMI, mean (SD), kg/m^2^	23.1 (3.5)	28.0 (2.6)	27.2 (3.1)	25.5 (3.4)	23.7 (3.2)	22.0 (2.8)	20.8 (2.3)	21.2 (2.1)	<0.01
Total cholesterol, mean (SD), mg/dL	192.2 (35.1)	231.9 (27.9)	221.5 (32.9)	209.6 (34.7)	198.9 (34.3)	187.2 (32.2)	171.5 (24.0)	171.2 (22.2)	<0.01
Fasting sugar, mean (SD), mg/dL	98.5 (18.6)	116.0 (29.8)	112.2 (29.7)	106.4 (24.7)	100.2 (18.3)	94.9 (12.4)	91.6 (8.0)	91.5 (7.3)	<0.01
Systolic pressure, mean (SD), mmHg	117.9 (16.7)	134.8 (12.5)	133.0 (14.7)	128.7 (15.6)	122.5 (15.8)	114.4 (14.6)	107.0 (11.1)	107.0 (10.4)	<0.01
Diastolic pressure, mean (SD), mmHg	71.1 (11.1)	83.1 (10.0)	81.0 (10.7)	77.7 (10.6)	73.7 (10.6)	68.9 (9.8)	64.6 (8.1)	64.7 (7.9)	<0.01
Creatinine, mean (SD), mg/dL	0.96 (0.20)	1.08 (0.16)	1.05 (0.18)	1.02 (0.19)	0.99 (0.21)	0.94 (0.19)	0.88 (0.18)	0.89 (0.18)	<0.01
**Categorical Variables ^b^**	**All** **(*n* = 169,366)**	**0** **(*n* = 2539)**	**1** **(*n* = 11,768)**	**2** **(*n* = 25,391)**	**3** **(*n* = 38,674)**	**4** **(*n* = 47,044)**	**5** **(*n* = 38,706)**	**6–7** **(*n* = 5244)**	***p*-Value**
Sex									<0.01
Male (%)	91,140 (53.8)	2466 (97.1)	9704 (82.5)	18,835 (74.2)	24,912 (64.4)	22,661 (48.2)	10,946 (28.3)	1616 (30.8)	
Female (%)	78,226 (46.2)	73 (2.9)	2064 (17.5)	6556 (25.8)	13,762 (35.6)	24,383 (51.8)	27,760 (71.7)	3628 (69.2)	
Education									<0.01
Below high school (%)	57,226 (34.3)	1024 (40.7)	5008 (43.2)	10,414 (41.6)	14,604 (38.3)	14,811 (31.9)	9604 (25.1)	1761 (34.0)	
beyond high school (%)	109,846 (65.7)	1489 (59.3)	6591 (56.8)	14,591 (58.4)	23,529 (61.7)	31,621 (68.1)	28,613 (74.9)	3412 (66.0)	
Family income									<0.01
<1.2 million NTD (%)	51,288 (76.1)	938 (73.1)	4122 (75.3)	8062 (74.6)	11,712 (75.6)	14,043 (76.8)	11,259 (77.5)	1152 (74.1)	
>1.2 million NTD (%)	16,142 (23.9)	346 (26.9)	1355 (24.7)	2747 (25.4)	3774 (24.4)	4245 (23.2)	3273 (22.5)	402 (25.9)	
Alcohol consumption									<0.01
Non-drinker (%)	134,006 (82.2)	1388 (56.0)	7559 (66.7)	17,954 (73.6)	29,605 (79.6)	38,526 (85.2)	34,425 (92.2)	4549 (90.3)	
Former drinker (%)	3976 (2.4)	130 (5.2)	504 (4.4)	877 (3.6)	988 (2.7)	991 (2.2)	402 (1.1)	84 (1.7)	
1–2 times/week (%)	16,820 (10.3)	562 (22.7)	1976 (17.4)	3606 (14.8)	4410 (11.9)	4042 (8.9)	1916 (5.1)	308 (6.1)	
3–4 times/week (%)	5501 (3.4)	268 (10.8)	843 (7.4)	1324 (5.4)	1451 (3.9)	1129 (2.5)	416 (1.1)	70 (1.4)	
≥5 times/week (%)	2665 (1.6)	130 (5.2)	454 (4.0)	634 (2.6)	715 (1.9)	528 (1.2)	176 (0.5)	28 (0.6)	
Blood pressure ^c^									<0.01
Not ideal (%)	74,827 (44.2)	2539 (100.0)	10,728 (91.2)	19,804 (78.0)	23,008 (59.5)	15,894 (33.8)	2620 (6.8)	234 (4.5)	
Ideal (%)	94,539 (55.8)	0	1040 (8.8)	5587 (22.0)	15,666 (40.5)	31,150 (66.2)	36,086 (93.2)	5010 (95.5)	
Total cholesterol ^d^									<0.01
Not ideal (%)	64,637 (38.2)	2539 (100.0)	9805 (83.3)	16,423 (64.7)	18,769 (48.5)	14,694 (31.2)	2227 (5.8)	180 (3.4)	
Ideal (%)	104,729 (61.8)	0	1963 (16.7)	8968 (35.3)	19,905 (51.5)	32,350 (68.8)	36,479 (94.2)	5064 (96.6)	
Fasting glucose ^e^									<0.01
Not ideal (%)	55,765 (32.9)	2539 (100.0)	9873 (83.9)	16,543 (65.2)	16,241 (42.0)	9043 (19.2)	1400 (3.6)	126 (2.4)	
Ideal (%)	113,601 (67.1)	0	1895 (16.1)	8848 (34.8)	22,433 (58.0)	38,001 (80.8)	37,306 (96.4)	5118 (97.6)	
Body mass index ^f^									<0.01
Not ideal (%)	45,136 (26.6)	2539 (100.0)	9908 (84.2)	14,706 (57.9)	11,914 (30.8)	5315 (11.3)	708 (1.8)	46 (0.9)	
Ideal (%)	124,230 (73.4)	0	1860 (15.8)	10,685 (42.1)	26,760 (69.2)	41,729 (88.7)	37,998 (98.2)	5198 (99.1)	
Smoking ^g^									<0.01
Not ideal (%)	45,765 (27.0)	2539 (100.0)	7266 (61.7)	11,272 (44.4)	13,180 (34.1)	10,243 (21.8)	1197 (3.1)	68 (1.3)	
Ideal (%)	123,601 (73.0)	0	4502 (38.3)	14,119 (55.6)	25,494 (65.9)	36,801 (78.2)	37,509 (96.9)	5176 (98.7)	
Physical activity ^h^									<0.01
Not ideal (%)	152,396 (90.0)	2539 (100.0)	11,456 (97.3)	23,761 (93.6)	35,266 (91.2)	42,421 (90.2)	34,482 (89.1)	2471 (47.1)	
Ideal (%)	16,970 (10.0)	0	312 (2.7)	1630 (6.4)	3408 (8.8)	4623 (9.8)	4224 (10.9)	2773 (52.9)	
Healthy diet score ^i^									<0.01
Not ideal (%)	155,000 (91.5)	2539 (100)	11,572 (98.3)	24,446 (96.3)	36,318 (93.9)	43,522 (92.5)	34,778 (89.9)	1825 (34.8)	
Ideal (%)	14,366 (8.5)	0	196 (1.7)	945 (3.7)	2356 (6.1)	3522 (7.5)	3928 (10.1)	3419 (65.2)	

^a^. Values in the continuous variables were expressed as mean (standard deviation) and analyzed by Student’s *t*-test. ^b^. Values in the categorical variables were expressed as number (%) and analyzed by the chi-square test. ^c^. Ideal blood pressure was defined as SBP < 120 mmHg and DBP < 80 mmHg. ^d^. Ideal total cholesterol was defined as <200 mg/dL. ^e^. Ideal fasting glucose was defined as <100 mg/dL. ^f^. Ideal BMI was defined as <25 kg/m^2^. ^g^. Ideal smoking status was defined as never smokers. ^h^. Ideal physical activity was defined as ≥210 min/week. ^i^. Ideal healthy diet scores was defined as 4–5 components.

**Table 3 nutrients-14-04040-t003:** Hazard ratio for the incident proteinuria between different CVH statuses and different number of ideal CVH metrics.

	Hazard Ratio (95% CI)		
	Unadjusted	Model 1 ^a^	Model 2 ^b^
CVH status (Number of ideal CVH metrics)			
Low CVH (0–2)	1 (ref)	1 (ref)	1 (ref)
Moderate CVH (3–4)	0.39 (0.35–0.44)	0.45 (0.40–0.50)	0.46 (0.37–0.57)
High CVH (5–7)	0.20 (0.17–0.24)	0.27 (0.23–0.32)	0.41 (0.30–0.55)
Number of ideal CVH metrics			
0	1 (ref)	1 (ref)	1 (ref)
1	0.64 (0.49–0.83)	0.60 (0.46–0.78)	0.73 (0.45–1.17)
2	0.40 (0.31–0.51)	0.38 (0.29–0.49)	0.50 (0.31–0.80)
3	0.26 (0.20–0.33)	0.26 (0.20–0.34)	0.34 (0.21–0.54)
4	0.15 (0.11–0.19)	0.16 (0.13–0.21)	0.22 (0.13–0.36)
5	0.10 (0.08–0.13)	0.13 (0.09–0.17)	0.25 (0.15–0.42)
6–7 ^c^	0.10 (0.06–0.15)	0.10 (0.06–0.16)	0.12 (0.04–0.36)

^a^. Model 1: adjust for age and gender. ^b^. Model 2: adjust for age, gender, education level, creatinine level, family income, and alcohol consumption. ^c^. Participants met 6 and 7 of ideal CVH metrics were calculated together due to small sample sizes.

**Table 4 nutrients-14-04040-t004:** Hazard ratio for incident proteinuria between each component of healthy diet score.

	Hazard Ratio (95% CI)		
	Unadjusted	Model 1 ^a^	Model 2 ^b^
Fruits and vegetables (≥450 g/day)	1.06 (0.95–1.18)	0.98 (0.88–1.10)	1.00 (0.82–1.22)
Fiber-rich whole grains (≥85 g/day)	1.33 (1.05–1.69)	1.04 (0.82–1.32)	0.93 (0.54–1.62)
Sodium (<1500 mg/day)	0.76 (0.63–0.91)	0.68 (0.57–0.82)	0.58 (0.43–0.79)
Fish (≥198 g/week)	1.10 (0.99–1.23)	1.02 (0.91–1.13)	1.03 (0.84–1.26)
Sugar-sweetened beverages (≤1 L/week)	1.24 (1.11–1.37)	0.97 (0.87–1.08)	0.88 (0.72–1.06)

^a^. Model 1: adjust for age and gender. ^b^. Model 2: adjust for age, gender education level, creatinine level, family income, and alcohol consumption.

## Data Availability

All the data were collected from the MJ Health Management Institution between 2000 and 2016 (http://www.mjhrf.org/main/page/resource/en/#resource08 (accessed on 4 August 2022)).

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
