# Peer review of "Relationship between Ideal Cardiovascular Health and Incident Proteinuria: A 5 Year Retrospective Cohort Study"

_nutrients, 2022, doi:10.3390/nu14194040_

Round 1
Reviewer 1 Report
Dear colleagues thank you very much for the submission
the following are suggestions to improve the paper
I suggest adding the table for the introduction that define what is ideal CVH per metric, this table is not only definition but also supported by literature from your region. I really don't understand why some metrics have such cut offs e.g., BMI was classified as ideal, intermediate, and poor according to values <25 kg/m2, 129 25-29.9 kg/m2, and ≥30 kg/m2, respectively.
Current table one is very congested I suggest making it into two tables - also try to make it more friendly by presenting it in landscape orientation
Since all data are available in continuous measure, I suggest using a z-score approach
Figure 2 is a repeat of the tables usually data should be presented once I suggest dropping the first plot because it's very difficult to follow specially with the small effect size and 95 confidence intervals
the discussion can be shortened and focused
some implications for practice need to be set as part of this study
I also suggest that you add a section about suggested work for future
Reviewer 2 Report
This is a retrospective cohort study regarding CVD metrics and proteinuria. The authors indicated a graded inverse relationship between the numbers of ideal CVH metrics and incident of proteinuria in a Taiwanese population. In particular, ideal blood pressure, ideal fasting glucose, and ideal BMI were found to be significantly protective against proteinuria.
This reviewer considers that the authors well performed the present study, and has some comments as described below.
Major comment:
1. Discussion section. In PubMed, a paper regarding CVH metrics and proteinuria has been published in 2022. The authors should discuss the similarities and differences in the Discussion section.
Minor comment:
2. Lines 356-359. Conclusion section seems to be in reference style. It should be corrected.
